# Study Replication: Shape Discrimination in a Conditioning Procedure on the Jumping Spider *Phidippus regius*

**DOI:** 10.3390/ani13142326

**Published:** 2023-07-17

**Authors:** Eleonora Mannino, Lucia Regolin, Enzo Moretto, Massimo De Agrò

**Affiliations:** 1Department of General Psychology, University of Padua, 35122 Padua, Italy; 2Esapolis’ Living Insects Museum of the Padua Province, 35143 Padua, Italy; 3Butterfly Arc Ltd., 35036 Padua, Italy; 4The BioRobotics Institute, Sant’Anna School of Advanced Studies, 56025 Pontedera, Italy

**Keywords:** amodal completion, Gestalt, vision, learning, invertebrates

## Abstract

**Simple Summary:**

Seemingly disconnected elements of the environment, like the two visible halves of an animal behind a tree, can be correctly interpreted as part of the same object. This process is referred to as “amodal completion” and seems to take place across profoundly different animal species. In a previous experiment, we tested the ability of jumping spiders to associate geometric shapes with sucrose rewards and then generalize the learned association to the shapes’ hidden versions. In that experiment, the spiders learned the association but failed the generalization task, leaving open the question of whether they are capable of amodally completing shapes. Here, we replicated the experiment, increasing the number of subjects and employing a deep neural network based scoring procedure. The results closely match those observed in the previous experiment, but without rising to significance. We stress the importance of employing hands-off approaches to scoring procedures, maximizing objectivity and efficiency.

**Abstract:**

Spiders possess a unique visual system, split into eight different eyes and divided into two fully independent visual pathways. This peculiar organization begs the question of how visual information is processed, and whether the classically recognized Gestalt rules of perception hold true. In a previous experiment, we tested the ability of jumping spiders to associate a geometrical shape with a reward (sucrose solution), and then to generalize the learned association to a partially occluded version of the shape. The occluded shape was presented together with a broken version of the same shape. The former should be perceived as a whole shape only in the case the animals, like humans, are able to amodally complete an object partly hidden by an occluder; otherwise, the two shapes would be perceived as identical. There, the spiders learned the association but failed to generalize. Here, we present a replication of the experiment, with an increased number of subjects, a DeepLabCut-based scoring procedure, and an improved statistical analysis. The results of the experiment follow closely the direction of the effects observed in the previous work but fail to rise to significance. We discuss the importance of study replication, and we especially highlight the use of automated scoring procedures to maximize objectivity in behavioral studies.

## 1. Introduction

The ability to recognize and categorize individual objects is critical for the survival of many animals. Although this is true for all the senses, vision has been the primary focus of such processes, in terms of how it allows creatures to pursue all major critical activities such feeding, predator avoidance, social interactions, and sexual behavior [1]. However, the visual scene is composed of a collection of seemingly unconnected elements and stimuli that need to be organized by the brain into meaningful units which are then recognized as objects. Gestalt psychology describes a set of rules according to which our minds organize and interpret visual data [2,3], under the assumption that the whole is greater than the sum of its parts. In other words, we do not simply focus on every separate component of the visual scene; instead, we aggregate multiple elements according to some similarities and perceive their global configuration. Such configuration is perceptually different from the array of elements composing it, in spite of the fact that the local information is identical in both cases.

An immediate application of such a system is apparent when considering the commonality of partially occluded figures. In the real world, objects present in the environment can often overlap: the ones closer to the observer can partially cover smaller or larger portions of the ones located farther away. Nevertheless, we remain capable of recognizing the individual fragments as a single object, as the visible parts are sufficient for completion of the contour. Amodal completion is the ability to perceive completed objects even in cases of partial occlusion [4,5], integrating fragmentary visual information from spatially segregated edges, and thus allowing the recognition of an object despite the presence of occluded parts. Gestalt psychology has described a set of perceptual rules that define which fragments are collected amodally into a single object and which are considered separate [2]. This mechanism has been widely studied in many species, including baboons [6], capuchin monkeys [7], chimpanzees [8,9], young chicks [10], pigeons [11], mice [12], and fish [13], and seems to be ubiquitous across all vertebrate taxa. This is probably due to the overarching need of animals to recognize meaningful objects from partial information, and thus essential for visually guided foraging or for identification of conspecifics and predators. However, even if this aggregation approach to object recognition seems to be quite successful, using the global configuration of the stimuli is not the only solution. Some animals do not seem to need the global information [9,14], favoring instead the analysis of specific local features [12,15], like specific colors or marking that can trigger recognition without the need for information on the global configuration.

The study of perceptual rules has also been extended to invertebrates, with mixed results. For example, the amodal completion mechanism has been observed to be present in cuttlefishes [16]. Among arthropods, bees can, for example, use different characteristics to detect and recognize flowers, such as shape [17], color or luminance contrast [18,19], pattern orientation [14], and symmetry [20]. All of these examples require the perception of the global configuration (i.e., the flower). On the other hand, social wasps can distinguish nest-mates based on subtle details such as facial markings [21], highlighting the importance of local cues. Similarly, spiders can discriminate between types of prey using local visual cues [22], while mantids use a “perceptual envelope” that includes size, contrast with the background, and location in the visual field to classify a stimulus as potential prey [23], all without requiring the “full picture”.

Salticids (jumping spiders) display some of the richest visually guided behaviors [24,25,26,27,28,29]. Their highly developed visual system is composed of four specialized and anatomically distinct pairs of eyes. The forward-facing pair of “anterior-medial” eyes is specialized in binocular and color vision [30,31,32], possesses a high visual acuity, and is characterized by a narrow (<5°) field of view [33,34]. The other three pairs of eyes, the “posterior-lateral”, “posterior-medial”, and “anterior-lateral” pairs, are positioned around the cephalothorax and grant the spider an almost 360° field of view [33]. These eyes are monochrome and have a lower visual acuity, but they can detect motion [35,36,37,38,39] thanks to their wide visual field and fast response time. Given these characteristics, jumping spiders have attracted the interest of scientists who wondered whether Gestalt principles still hold true in a visual system organized to separately detect different elements of the visual scene. For example, Dolev and Nelson [40] found that the jumping spider *Evarcha culicivora*, which is specialized in hunting mosquitoes, can recognize its preferred prey based on simple “stick figures” of it. Crucially, the spiders seemed to use the orientation of the body elements (abdomen, legs) relative to each other, suggesting the need for a representation of the whole body to allow recognition. However, the spiders continued to recognize images as prey if the position of the elements was scrambled, suggesting a focus on the local characteristics. It is crucial to point out, however, that this level of local prominence may be dependent on the predatory specialization of *E. culicivora*. When testing *Hypoblemum albovittatum*, a generalist predator, the authors found a general preference for a realistic type of image [41]. This species seemed to rely on a more holistic approach than *E. culicivora*, lacking preferential attention to specific figure elements. In another experiment, Rößler et al. [42] presented *Salticus scenicus* with 3D models of bigger jumping spiders, which caused the subjects to run away after an initial freezing. When presented with an equally sized 3D blob, *Salticus scenicus* did not react. Curiously, if the spiders were presented with a 3D blob with spider eyes pasted on it, the subjects had a mixed reaction, with some of them freezing and escaping, while others did not respond. This suggests that while the local feature of the eyes is relevant, this is not always sufficient, requiring the whole figure to trigger recognition. As it stands, the validity of Gestalt principles in jumping spiders remains unclear.

In a previous study [43], we trained the jumping spider *Phidippus regius* to discriminate among two geometrical shapes, a circle and a cross. Afterwards, we asked the animals to choose between two versions of the rewarded shape: one partially occluded, and one cut in sections. If jumping spiders were able to amodally complete the missing parts, we should have observed a generalization of the learned association towards the occluded version. Conditioning was effective, as spiders were able to learn the discrimination task. However, they did not generalize the association to the illusory stimulus, suggesting that spiders’ visual system may not distinguish between the two configurations present at test. It may well be that jumping spiders do not perform amodal completion. However, it is difficult to take these results at face value, due to some shortcomings of the previous study. First, the experiment included only 18 subjects, which, coupled with the low response rate of the experiment, significantly lowered the statistical power of the analysis. Even more crucially, the spiders’ choices were scored manually by the experimenters, who had to judge from the videos whether the spiders drank the presented reward. This left the decision of which trials to include in the analysis to the experimenters. This can introduce important subconscious biases in the scoring: even though a double-blind procedure was implemented, an experienced scorer could pick up subtle elements of the spiders’ behavior suggesting the experimental condition, as well as whether the food contacted by the spider was a reward or not. Especially given the low number of subjects, the experimenter’s decisions of counting or not a behavior as a choice in a handful of trials had a strong weight on the final results [44].

In the current study, we replicated fully the experiment presented in De Agrò et al. [43]. However, we increased the number of subjects, as well as implementing an automated scoring procedure based on the pose-estimation software DeepLabCut v2.2b8 [45]. This was undertaken to completely remove the experimenters from the processes of data inclusion decision-making and video scoring, so as to guarantee maximum objectivity.

## 2. Materials and Methods

A new group of jumping spiders underwent exactly the same experimental procedure described in De Agrò et al. [43]. However, the scoring and analysis procedures are completely new.

### 2.1. Subjects

Thirty *Phidippus regius* were used in this experiment. They were all bred in our lab within a month of each other, stemming from 5 different egg sacs from 3 different parent pairs. Spiders were kept individually in transparent plastic boxes (7 × 16 × 6 cm); rearing boxes were enriched with a piece of a cardboard egg carton to allow spiders to build nests and hide, and a water dispenser that was topped up as needed. During standard maintenance, spiders were fed *ad libitum* with *Tenebrio molitor* larvae. This feeding routine was interrupted 3 days before the start of the testing period to ensure motivation. Furthermore, they received one *Tenebrio molitor* larvae every week during the resting periods of the procedure (see below). Lighting in the laboratory consisted of neon lamps with natural light (5000 kelvin color temperature, 36 watts, 3350 lumens) on a 12 h light/dark cycle with lights on at 08:00 h. Subjects were well-acclimated to the laboratory before testing and had no previous experience with the types of stimuli presented. The experiment was performed around 4 months after hatching, when some of the spiders reached sexual maturity. Tests were carried out between 09:00 and 18:00 h.

### 2.2. Apparatus

Trials were performed in plastic boxes of the same size as the housing boxes. The walls of the boxes were lined from the outside with white paper to preventing the subjects viewing other spiders and the lab during tests, while the lid remained transparent to allow the light to pass through and to allow the camera to record (Figure 1A). Two L-shaped white plastic blocks (Figure 1B) were set in each box and two red-colored drops (≈40 µL) were placed behind them so that spiders could not see them from their starting visual perspective. As in the previous study, the vertical wall of the white blocks showed either an ‘O’ or an ‘X’, both colored in black and matched for total area (7 mm^2^). Each drop contained sugar (20%) (as positive reward) or citric acid (25%) (as punishment) and they were visually identical; however, since sugar absorbs ultraviolet light [46], spiders could have perceived the color of the drops differently, creating the need for the vertical plastic screen hiding the drop from sight from the perspective of the starting position.

### 2.3. Design and Procedure

Each individual was subjected to a training phase and a testing phase. To ensure a high level of motivation and to maximize responsiveness, spiders were starved for 3 days prior to testing. The day before testing, spiders were placed in new empty boxes that were identical to the housing ones, where they were kept for the entire duration of the experiment.

Every subject performed 3 trials a day for 6 days, with a 1 h long interval between trials to avoid excessive stress. At the start of each trial, spiders were carefully placed in the center of the box, while the L-shaped blocks with the red-colored drops were positioned at the short sides of the box. Thus, the spiders could easily see the figures from their from the starting position, but could not see the drops. Once in place, the spider was given 45 min to freely move around the box and had free access to the water drops. The shapes were randomly assigned, and we kept the combination of figure and sugar or citric acid constant for each subject. To ensure that spiders learned to associate the figure with the drop flavor, not the location, the location of drops was changed semi-randomly trial by trial. Moreover, the boxes were always rotated in a different orientation for every trial, to ensure the absence of any external or internal visual landmark.

The testing procedure lasted for 2 weeks and took place 2 days after the training phase. Each subject carried out 18 trials, divided between 12 “rewarded trials” (RT), 3 “unrewarded shape” trials (US) and 3 “unrewarded illusion” trials (UI). The RT were 2/3 of these 18 trials, and the same stimuli as in the training phase were used. These were included to maintain the supposed learned association. The remaining number of trials were split between US, where the stimuli were presented without the reward/punishment, and UI, where the spiders were tested with a completely new stimulus, still without the presence of reward.

### 2.4. Visual Stimuli

During the course of experiments, we used a total of 6 different visual stimuli, glued at the front and back of the vertical wall of the L-shaped blocks. In RT and US trials, the two shapes were an “X” and an “O”. Both were black in color and covered roughly the same total area (7 mm^2^). Half of the spiders always received a reward associated with the “X”, and the other half with the “O”. By contrast, in US, trials we presented to the spiders two versions of the same shape the spider had been trained on. The first option was an “occluded” version of the trained shape: the “X” or the “O” were placed behind a red horizontal bar. If spiders possess the mechanism of amodal completion, they should perceive the shape behind the bar as whole, and, as such, identical to the trained shape. The second option was a “cut” version of the shape, where we removed a horizontal portion of the figure, as wide as the red bar in the occluded version. In this case, no occluder hid the missing portion, and as such, amodal completion should not induce the perception of a whole shape; instead, it should be perceived as a novel stimulus. To maintain the same low-level stimuli characteristics, we included the red bar in the stimulus, placing it below the cut shape (Figure 1C). The bar was colored differently from the shape to ensure perceptually noticeable difference. Jumping spiders do not have a red-light photoreceptor. However, it has been demonstrated that at least one species can perceive red thanks to a pigment layer in the retina [32]), and were shown as able to perceive this color in foraging contexts [31,47]). *Phidippus* spiders often possess red-colored markings. In any case, regardless of its color, the luminance of the red bar was different from the one of the shape, which should make the difference between the two perceivable.

### 2.5. Scoring and Data Analysis

To record the experiment, we placed 4 boxes near each other, 60 cm below a Raspberry Pi equipped with Raspberry Pi digital cameras (V1.3). Each camera recorded at 5 fps and 1080 p resolution. Using a total of 4 identical setups, we were able to record 16 spiders at once.

Afterwards, we used the Python software DeepLabCut [45] to train a neural network capable of recognizing 4 points on the spiders’ bodies (position of the two ALEs, pedicel, and spinneret), as well as the position of the two drops. We then fed all the recorded videos to the trained network, which outputted the frame-by-frame coordinates of the trained points. Subsequently, using a custom script written in Python 3.10 [48], using the packages NumPy 1.24.3 [49,50], SciPy 1.10.1 [51], and pandas 2.0.0 [52], we recorded all events where the two ALE points contacted the drop (meaning they reached a distance from the drop center <5 mm, observed to be the average drop radius). These were considered potential choice events. All these occurrences were then watched by an experimenter, who manually removed false positives. This was needed mostly because DeepLabCut was unable to discriminate spiders walking on the box lid from spiders walking on the ground. As such, it recorded as contact events many occurrences where the animals simply walked on the ceiling above the drops. In this phase, the experimenter also removed other false positives, like events where the animal did not cross the selection area or just sprinted across. Crucially, in order to minimize bias, the videos were shown to the experimenter without an indication of the video condition, subject, or date. All the remaining events were kept for the analysis. For every spider, we recorded the number of visits to each drop in each trial. Many spiders only contacted one of the two drops for the whole trial duration: in those cases, we also recorded a binomial choice, with 1 being only contacts with the rewarded shape, and 0 being only contacts with the punished one.

All the analyses were performed in R4.1.2 [53], using the libraries glmmTMB 1.1.7 [54], car 3.1-2 [55], DHARMa 0.4.6 [56], emmeans 1.8.6 [57], ggplot2 3.4.2 [58], and reticulate 1.28 [59]. Plots were generated using Python and the packages seaborn 0.12.2 [60] and matplotlib 3.7.1 [61]. We modelled binomial choice using generalized mixed-effect models with a binomial error structure. We included in the model both the trial type and trial number and included the trial type nested in subject identity as a random intercept. After evaluating the goodness of fit for the model, we examined the effect of the predictors using an analysis of deviance, and subsequently performed Bonferroni-corrected post hoc analysis on factors that had a significant effect. We performed a second analysis on the number of contacts to each option in each trial, using a generalized mixed-effect model with a Poisson error structure. We included in the model the visit value (rewarded or unrewarded shape), the trial number, and the trial type. We again included the trial type nested in subject identity as a random intercept. After again evaluating the goodness of fit and observing the effect with the analysis of deviance, we performed Bonferroni-corrected post hoc analysis on relevant factors.

## 3. Results

Raw data for the experiment are available in Appendix A. For the full analysis and script, see Appendix A. Only the most relevant results are reported below. All the tests were performed on weeks 2 and 3 of the experiment, as week 1 only included training trials. We observed a generally low response rate: spiders contacted any one of the two drops in only 7.4% of the RT trials, 9.1% of the US trials, and 17.8% of the UI ones. The latter appeared to be a significantly higher proportion with respect to RT (post hoc, Bonferroni corrected, odds. Ratio = 2.727, SE = 0.725, DF = 1413, *t* = 3.771, *p* = 0.0005, *n* = 30).

Considering the binomial choice, we observed a significant effect of trial number (GLMM analysis of deviance, χ^2^ = 6.0311, DF = 1, *p* = 0.0140, *n* = 30), but no effect of trial type (χ^2^ = 3.958, DF = 2, *p* = 0.18307, *n* = 30), nor of the interaction between the two (χ^2^ = 0.9012, DF = 2, *p* = 0.63725, *n* = 30). The post hoc analysis revealed that spiders chose the correct shape significantly more often in the rewarded trials (post hoc Bonferroni corrected, prob. = 0.786, SE = 0.058, DF = 106, *t* = 3.771, *p* = 0.0008, *n* = 30), but performed at chance level in the unrewarded-shape trials (prob. = 0.664, SE = 0.113, DF = 106, *t* = 1.350, *p* = 0.5394, *n* = 30) and the unrewarded-illusion trials (prob. = 0.578, SE = 0.095, DF = 106, *t* = 0.806, *p* = 1, *n* = 30) (Figure 2A). We also observed a nonsignificant increase in correct choices across subsequent trials, independent of trial type (trend = 0.0542, SE = 0.0277, DF = 106, *t* = 1.957, *p* = 0.0530, *n* = 30).

The model performed using the number of contacts to each option gave very similar results. We observed a significant effect of the visit value (whether it was to the rewarded or to the punished option. GLMM analysis of deviance, χ^2^ =21.73, DF = 1, *p* = 0.0001, *n* = 30). We also observed an effect of trial type (χ^2^ =9.6217, DF = 2, *p* = 0.008, *n* = 30), of trial number (χ^2^ = 6.3727, DF = 1, *p* = 0.0116, *n* = 30), of the interaction between visit value and trial type (χ^2^ = 10.459, DF = 2, *p* = 0.0054, *n* = 30), and of the interaction between visit value and trial number (χ^2^ = 12.1545, DF = 1, *p* = 0.0005, *n* = 30). The post hoc revealed that spiders visited significantly more often the correct option over the incorrect one in rewarded trials (post hoc, Bonferroni corrected; ratio = 5.57, SE = 1.68, DF = 2271, *t* = 5.696, *p* < 0.0001, *n* = 30), but did not so in unrewarded-shape trials (ratio = 2.56, SE = 1.265, DF = 2271, *t* = 1.909, *p* = 0.1693, *n* = 30) nor in unrewarded-illusion trials (ratio = 1.49, SE = 0.456, DF = 2271, *t* = 1.288, *p* = 0.5938, *n* = 30). Regarding the trend across trials, we observed that the spiders contacted the wrong option less across trials for the rewarded trials (trend = −0.082, SE = 0.0241, DF = 2271, *t* = −3.405, *p* = 0.004, *n* = 30) and for the unrewarded-illusion trials (trend = −0.06, SE = 0.0224, DF = 2271, *t* = −2.684, *p* = 0.0439, *n* = 30) but not for the unrewarded-shape trials (trend = −0.02101, SE = 0.0419, DF = 2271, *t* = −0.501, *p* = 1, *n* = 30) (Figure 2B). No trend was appreciable for the number of contacts with the correct option (trends: RT = −0.00817, US = −0.0049, UI = −0.0045, all *p* values > 0.05, *n* = 30).

## 4. Discussion

In this study, we replicated the experiment by De Agrò et al. [43], using a new, larger sample of spiders, and we implemented a fully automated scoring procedure.

The spiders performed significantly above chance level in the rewarded-shape condition, both in terms of the percentage of correct trials (78.6% of choices for sucrose solution drop) and the raw number of contacts to either drop (correct drop touched 5.57 times more than the incorrect one). This is consistent with what was previously observed [43]. The percentage of correct choices is far lower than the one observed in our previous study (96.8%). This is probably caused by the automatic scoring procedure: DeepLabCut introduces a certain level of noise that can cause false positives. On the other hand, incorrect events that may have been discarded during manual scoring due to experimenter bias are correctly included here. Regardless of the difference, this result confirms that the training procedure is an effective motivational tool for jumping spiders: all individuals consistently avoided the citric acid solution while approaching and consuming the sucrose one. It is crucial to point out, however, that performance in the rewarded-shape trials does not equate to a successful association between the reward and the conditioned stimulus. Indeed, the spiders could rely on other cues to avoid the citric acid solution, including drop scent or subtle color differences. These, even if invisible to the human eye, could be evident from the spiders’ sensorium [30,46,62].

In the unrewarded-shape condition, the spiders preferred the correct shape 66.4% of the time, but this preference was not significantly different from chance. Similarly, when looking at the raw number of visits, the spiders contacted the drop by the trained shape 2.56 times more, but, again, this was not significantly different from chance. In our previous study, [43], spiders preferred the trained shape 70.9% of the time, which is very similar to the findings of the current study. The consistent effect direction across the two experiments seems to suggest that the spiders learned the task, but the overall low response rate decreased the statistical power of the analysis. On the other hand, it is possible that we previously committed a type I error, declaring the spiders capable of learning the task when instead their performance was due to chance. It is impossible with the available data to tell which of the two options is correct.

In the unrewarded-illusion condition, the spiders preferred the occluded shape 57.8% of the times, a nonsignificant percentage. Similarly, when looking at the raw number of visits, the spiders contacted the drop by the occluded version of the shape 1.49 times more than the broken one, again not significantly different from chance. This is similar to what we previously observed, as in De Agrò et al. [43]: the spiders chose the occluded shape exactly 50% of the time. In the absence of a significant result in the unrewarded-shape condition, we cannot make any inferences about the amodal completion skills of jumping spiders. Indeed, if the spiders did not associate the shape with a reward, both options in the shape-illusion condition would have no value, even if correctly perceived.

We believe that the DeepLabCut-based automatic scoring procedure that we implemented is an enormous improvement to the procedure. This allowed us to increase scoring objectivity by minimizing human involvement. It should be noted, however, that human involvement is not fully removed from the procedure, and it could never be: A human must train the network, a human-specified criterion for event detection had to be employed, and a human experimenter had to confirm the events. As such, biases could not be completely removed, only decreased. Excluding the objectivity benefits, we also observed a significant decrease in scoring time and a virtual zeroing of person-hours invested. In this experiment, we collected 1425 videos for a total of c. 1068 h of footage, which we would have previously rewatched fully to detect events. This greatly limits the scalability of the experiment, as adding a single subject increases the footage to score by 40 h. DeepLabCut instead requires only an initial investment in training the neural network of around 5 h, independent of the number of subjects. Thereafter, scoring time becomes negligible, and crucially requires no involvement by a human. The procedure comes with its own shortcomings, however. Mainly, it introduces a variable amount of noise in the data, which will in turn require a higher sample size to be averaged out. This, coupled with the low response rate of this behavioral procedure, makes it very difficult to observe any significant results. We are persuaded that the benefits offered by automation greatly outweigh the costs. However, nowadays a lot of established and reliable procedures are available to study both learning [24,63,64,65] and visual abilities [25,26,39,66,67] in jumping spiders, making the one described here often a suboptimal solution.

## 5. Conclusions

In this paper, we trained jumping spiders in a shape discrimination task. Then, we asked whether they were capable of generalizing the learned association to occluded versions of the trained shape, demonstrating the presence of amodal completion abilities. The experiment was designed as a replication of De Agrò et al. [43]. The results of this replica seemed similar to the one observed in the original study, but ultimately failed to rise to significance, leaving whether amodal completion is present in these animals an open question. We hope that future research will attempt to close this knowledge gap, as understanding whether Gestalt principles hold true in a modular visual system could bring surprising new insights. Indeed, spiders live in a world full of occluding elements, mimetic predators and prey, and varied backgrounds. As such, the ability to perceive complete objects in this complicated environment should be important to their survival. If jumping spiders truly do not rely on the mechanism of amodal completion, we must assume that they apply a yet unknown, but different, solution to the problem.

## Figures and Tables

**Figure 1 animals-13-02326-f001:**
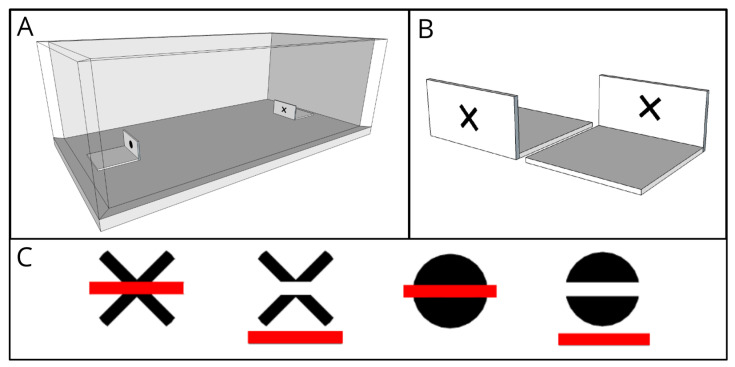
(**A**) Experimental apparatus (7 × 16 × 6 cm) with the two platforms, X and O, located at both the short-side ends of the arena. Each spider was placed in the center of the arena and had 45 min to freely explore and choose to approach either figure. (**B**) Example of the “X” platform where the drop (40 µL) was placed; front and back views. (**C**) Stimuli employed in the amodal completion test. From the left, spiders trained on the “X” shape were exposed to the first vs. second stimulus, while spiders trained on the “O” shape were exposed to the third vs. fourth stimulus. Each pair of stimuli are identical with regard to the local features of the black area, but differ in their global configuration (to the human eye) (figures from [43]).

**Figure 2 animals-13-02326-f002:**
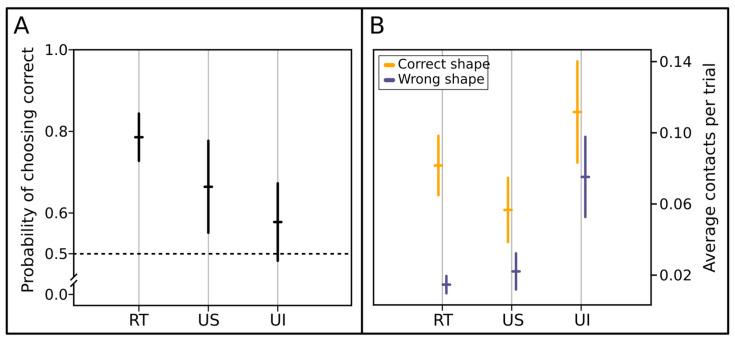
Results. (**A**) Binomial choice. On the *X* axis is reported the trial type. On the *Y* axis, the probability of choosing the shape associated with sugar water is shown. Error bars represent standard error. *n* = 30. Spiders significantly preferred the rewarded shape in rewarded trials (RT, prob. = 0.786, *p* = 0.0008) but performed at chance level in the unrewarded-shape trials (US, prob. = 0.664, *p* = 0.5394) and unrewarded-illusion trials (UI, prob. = 0.578, *p* = 1). (**B**) Contacts frequency to each drop. The *X* axis shows the trial type. On the *Y* axis, the average contacts per trial is shown. Error bars represent standard error. *n* = 30. Spiders significantly visited more often the correct option over the incorrect one in rewarded trials (RT, ratio = 5.57, *p* < 0.0001), but they did not in unrewarded-shape trials (US, ratio = 2.56, *p* = 0.1693), nor in unrewarded-illusion trials (UI, ratio = 1.49, *p* = 0.5938).

## Data Availability

Raw data are included as a Appendix A.

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
