# Peer review of "Study Replication: Shape Discrimination in a Conditioning Procedure on the Jumping Spider Phidippus regius"

_animals, 2023, doi:10.3390/ani13142326_

Round 1
Reviewer 1 Report
The paper repeats, with a small modification in the method, the previous experiments by De Argo et al, 2017. In the previous study the authors failed to reject the null hypothesis that the jumping spiders do not generalise to illusory stimulus. However, in the introduction the authors incorrectly interpret the results of the previous study. They claim that jumping spiders tested in the previous study did not generalise to illusory stimulus. This is a logical error. The conclusion from previous study should be that the data collected in the previous study does not allow us to judge whether the spiders able to see illusory contours or not able to see illusory contours because the sample size was not sufficiently large. In the present study, the null hypothesis also was not rejected. Therefore, no definite conclusion can be made. However, there is a tendency in the results. Hence a study with a larger sample size is likely to establish that jumping spides see illusory contours. I suggest that the authors estimate the sample size needed to reach significance. I understand that the experiments are time consuming. However it would be good to increase the number of animals tested.
The stimuli design: the authors use a red stripe to occlude the shape. However, if spiders are not sensitive in the red part of the spectrum, for spiders, the red stripe would not be different from a black. In this case, the patterns do not look as occluded cross or circle, but rather as novel pattern. I suggest that a different colour of the stripe is used in future experiments. Also it would be good to include in this manuscript a discussion of the spectral range of spider’s vision.
Minor comments.
Discussion need to be rewritten in such way that the absence of statistical significance is not equated with the absence of effect (null hypothesis cannot be rejected but cannot be accepted)
Add description of stimuli as a separate subsection to the methods section. This should be done in addition to the figure legend.
When describing statistical significance, add sample size in brackets. Also, add to the figure legend the sample size.
Lines 55-56 – Incomprehensible.
Lines 59-62 – Reword and check grammar.
Lines 64-65 – Change “their” to “invertebrates.” Note clear what is meant.
Add description of stimuli as a separate subsection to the methods section. This should be done in addition to the figure legend.
309 – Incorrect to say that the results are ‘in contrast’. Comparing the levels of significance between two studies is an invalid procedure. Delete this phrase and go to the next argument.
317 – Type one error – is finding the difference when there is no difference. It is not finding ‘significant difference’ while there is ‘no significant difference’.
320 – Accepting null hypothesis is a logical error. You can only conclude that you filed to reject the null hypothesis and hence we do not know if the null hypothesis is correct or not.
Lines 55-56 – Incomprehensible.
Lines 59-62 – Reword and check grammar.
Lines 64-65 – Change “their” to “invertebrates.” Note clear what is meant.
I suggest that a native speaker reads the ms.
Author Response
Please see the attachment containing the response to both reviewers

Reviewer 2 Report
The manuscript by Mannino and colleagues describes an interesting study on jumping spiders’ visual ecology. The study aims at resolving a question of whether Gestalt principles (specifically amodal completion) work in animals possessing modular visual system. The jumping spider system studied by the authors, in which particular elements separately detect different objects of the visual scene, seems quite unique in the context of perceiving and interpreting signals and the general issue is important for visual ecology studies. It does not only tell how the particular animal group perceives partially occluded objects, but also, how in the visual and cognitive systems the problems of identification and interpretation of specific visual signals are solved.
The manuscript describes a replication of a former study of one of the authors (De Agro et al. 2017). This time the study is based on a higher number of subjects and a different method of scoring behavioural data. Authors used standard methods of both, animal maintenance and testing and put a lot of effort into experimental procedures to make the results more objective and to raise the power of statistical tests. They increased a number of subjects to 30, which resulted in raising the level of significance and also used automatic scoring procedures applying artificial neural networks. The study is carefully prepared, carried out and the whole manuscript is clear and well-written. I find it an interesting contribution to understanding, how jumping spiders perceive and interpret visual signals.
In addition to the overall positive opinion on the manuscript, I have a few comments:
Object (specifically prey) identification abilities are highly dependent on predator’s specialization. In jumping spiders this is well-studied in generalist and specialist jumping spiders. In the introduction (P2, L74), however, it is only touched upon with one example. I suggest considering a bit broader presentation of the problem based on more recent studies.
The first line of paragraph 2.1. (Subjects) describes the animals used in the experiments. The statement: “Thirty between juveniles and adults Phidippus regius were born in captivity…” (P 3, L 137), seems, however, too vague. As the age and experience in jumping spiders may affect both, their visual perception and their decisions based on the perceived signals (e.g. Nelson, Jackson & Sune 2005; Bartos & Szczepko 2012) it seems crucial to at least mention age groups used in the experiments (defined by the number of days after hatching or the number of their moults or their size). It should also be added, whether the spiders, which “hatched in captivity”, originated from a wild population (brought directly from North America?) or from population/s bred in captivity for a long time. If the latter the case, which “variety” was used in the tests. All these characteristics may affect spider behaviour and should be described in the methods.
Author Response
Please see the attachment containing response to both reviewers

Round 2
Reviewer 1 Report
I am satisfied with corrections of the manuscript.
Author Response
Thank you for your inputs!